# Recent Advances in Handedness Genetics

**Silvia Paracchini** [ORCID]

School of Medicine, University of St Andrews, North Haugh, St Andrews KY16 9TF, UK; sp58@st-andrews.ac.uk

**Abstract:** Around the world, about 10% people prefer using their left-hand. What leads to this fixed proportion across populations and what determines left versus right preference at an individual level is far from being established. Genetic studies are a tool to answer these questions. Analysis in twins and family show that about 25% of handedness variance is due to genetics. In spite of very large cohorts, only a small fraction of this genetic component can be pinpoint to specific genes. Some of the genetic associations identified so far provide evidence for shared biology contributing to both handedness and cerebral asymmetries. In addition, they demonstrate that handedness is a highly polygenic trait. Typically, handedness is measured as the preferred hand for writing. This is a very convenient measure, especially to reach large sample sizes, but quantitative measures might capture different handedness dimensions and be better suited for genetic analyses. This paper reviews the latest findings from molecular genetic studies as well as the implications of using different ways of assessing handedness.

**Keywords:** handedness; neurodevelopment; GWAS; heritability; quantitative trait; polygenic scores

## 1. Is Handedness a Genetic Trait?

Before embarking in the search for the genetics factors of any traits, the most fundamental question is whether a trait is influenced by a genetic component. More specifically, we are asking whether the variability observed in the population for that particular trait is influenced by genetics. Several observations confirm that a genetic component contributes to handedness.

Most people can readily say whether they are left- or right-handed, especially for highly skilled task like writing with a pen. Of course, it is possible to learn writing with the nonpreferred hand, but at least at the beginning, that would feel an un-natural act. Based on these observations we can state that it is in our nature to have a preferred hand for writing, which is the right hand for most people. Probably because of the minority status, left-handers were stigmatised throughout history and cultures. In fact, it is quite common to hear of left-handers being forced to use their right hand for some tasks such as writing. Instead, the reverse, i.e., forcing right-handers to use the left hand, is very unlikely. This phenomenon is well-documented in the UK Biobank data showing that the prevalence of left-handedness increases in younger participants probably because of stronger stigma in older generations [1]. A recent meta-analysis, confirmed the same historical trend and that left-handedness tends to converge to around 10% across populations [2]. Although left-handedness prevalence tends to remain low in some countries, e.g., China, this seems to be a cultural effect. For example, a 1980s survey reported that less than 1% Chinese students are left-handed [3]. A more recent study, reported a higher prevalence of left-handedness (6%) in a Chinese cohort living in Hong Kong, possibly as a result of the westernisation of this region [4]. Therefore, left-handedness, not only is a minority status, but appears to be fixed to a constant frequency. This fixed prevalence is suggestive of evolutionary forces maintaining the ratio of 1 left- to 9 right-handers possibly through genetic mechanisms. This scenario could be explained by a frequency dependent selection process where the minor trait has an advantage but only until it remains at low prevalence

in the population [5–7]. A cost is clearly associated to left-handedness, else we would observe it at a 50% frequency in populations.

The link between handedness and language is another indicator of the biological nature of handedness. Although both hemispheres are engaged during language tasks, for the majority of people, hemispheric dominance resides in the left side. Right hemisphere dominance for language is rare and observed preferentially in left-handers [8] (see also Corballis [9] and Vingerhoets et al. [10] in this issue for details on functional and anatomical brain asymmetries). This link is weak but suggests some common pathways control the establishment of brain asymmetries and contribute to both language and handedness.

Family and twin studies provide the most compelling case in support of genetics, indicating that at least one quarter of handedness variance is determined by genetic factors [11]. However, the remaining 75% are not necessarily influenced by nongenetic or environmental factors. For example, intrinsic variability linked to developmental processes might explain a large of portion of the remaining variability across people, as argued by Kevin Mitchell [12] and, more recently, by Chris McManus [13], as part of a discussion setting the vision for the future of laterality research [14,15]. The idea is that, while the general developmental stages of an individual are directed by biological processes tightly regulated by our genes, a random component allows fluctuations from the general plan. Such fluctuations, which are actually part of the biological plan itself, could play an important role in determining an individual's characteristics, including handedness. Under this view, the actual genetic component of handedness is expected to be much higher than what (~25%) predicted by twin studies. McManus' prediction is that very few environment factors are likely to play any significant role in establishing the direction of hand preference.

## 2. How to Measure Handedness

Having established a firm and conspicuous genetic component underlying a trait, the next question is how best to measure the phenotype for genetic analyses. Handedness appears to be a very straightforward phenotype, with most people being able to define themselves as either left- or right-handed, typically on the basis of their preferred hand for writing. The majority of individuals also carry out other tasks preferentially with the same hand they used for writing, either the left or the right one. However, a minority, defined as mixed-handed, prefer using different hands for different activities (e.g., writing with the right but throwing a ball with the left hand) and a small group, or ambidextrous, has no clear hand preference between the two hands. In total, mixed-handed and ambidextrous individuals are about 9% of the population, a group almost as big as the left-handers [2]. Tools like the Edinburgh Handedness Inventory (EHI) and Annett's questionnaire [16,17], which record the preferred hands for a dozen of activities or items, allows identifying these individuals. While most people will answer "right" or "left" for all items, there will be a group without consistent preferences. Instead, one task alone, e.g., hand preference for writing, cannot identify this group. A third possibility is to measure handedness as relative hand skills by assessing how better one individual performs with one hand versus the other. This approach leads to continuous measures, or laterality quotients (LQ; Figure 1). The pegboard task, which records the time taken to move pegs in a row of holes [17], is a commonly used tool to derive such scores. A key question is whether different handedness measures, which require significant time or resources to be collected in large cohorts, offer any specific advantage for genetic studies over the self-reported measure of hand preference for writing [18].

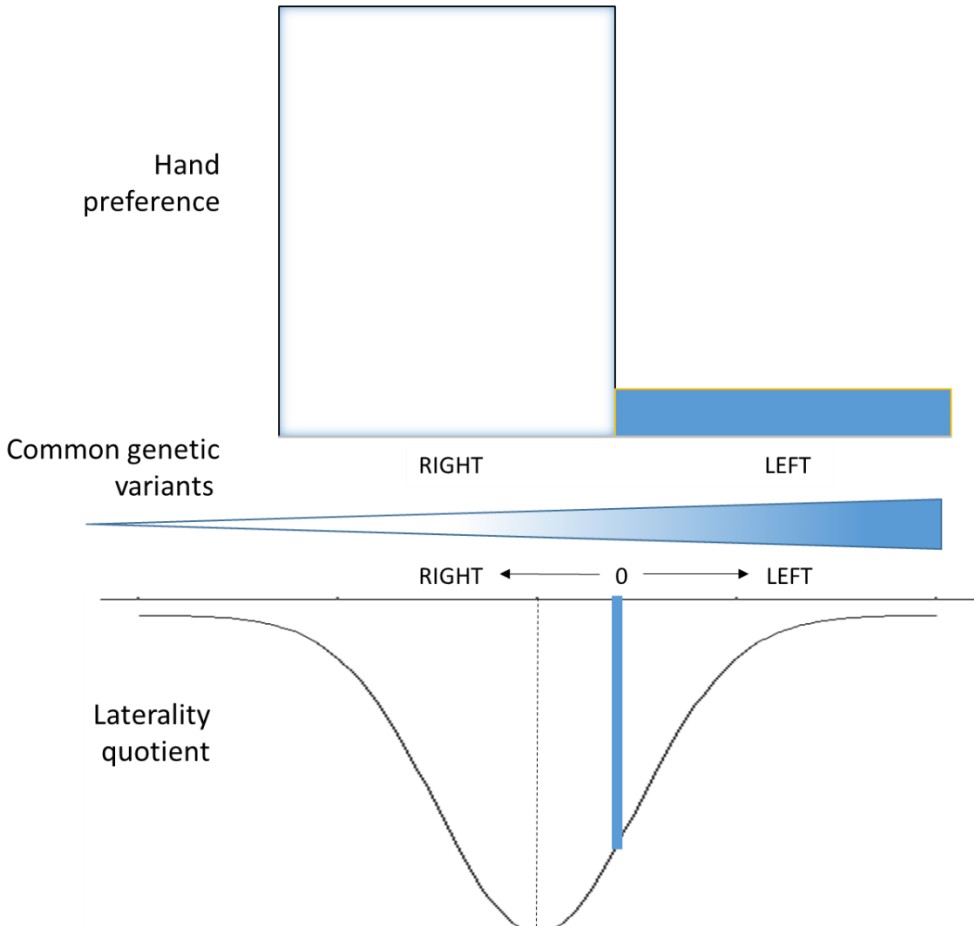

**Figure 1. Polygenic model for handedness.** Handedness is typically measured as hand preference (top bars). But it can also be measured along a continuum using laterality quotient (bottom curve, shown upside-down for convenience). Hand preference leads to two categories: right and left distributed in a 9:1 ratio. Laterality quotients (LQ) assess relative hand skills and how much an individual is lateralised in addition to a left v right direction. A value of zero (0, aligned along blue line) indicates equal ability with both hands and separates left and right handers for that particular skill. Different LQ identify a general left v right component, but do not correlate perfectly with hand preference. The chance of being left-handed increases with accumulation of multiple genetic variants represented by gradient in middle of figure. Poor correlation across handedness measures suggest that different pools of common variants contribute to different measures, although we expect some overlaps. For example, different genetic studies reported associations with different set of genes with cytoskeletal functions. Although hand preference is a convenient measure, which can be easily collected in very large cohorts, LQ might be better suited to identify genetics underlying handedness.

A starting point to address this issue is to examine how different measures correlate with each other and whether different types of assessment can be used interchangeably. This can be done in population-based cohorts that include thousands of participants characterised with multiple handedness measures. For example, participants of the Avon Longitudinal Study of Parents and Children (ALSPAC) cohort [19] were assessed with hand preference at different time points, handedness questionnaires, and multiple motor tasks, which can be used to derive LQ (N up to 8000). Thanks to these data, we showed that different laterality measures are poorly correlated with one another [20] and, beyond capturing a general left/right component, they tap in different laterality dimensions. Moderate correlation (0.42) for handedness measures derived from the EHI and the pegboard task was also reported in 205 twin pairs recruited in Hong Kong. Remarkably, both measures presented similar heritability estimates at around 20%, but the low correlation suggest that most likely they are underpinned by different genetic factors [4].

The UK Biobank [21] with its multilayers of biological, genetic, clinical, and behavioural data for half million study participants is revolutionising different lines of research, including in the field of laterality, as we will discuss later (see also Corballis in this issue [9]). However, in this cohort the handedness assessment is limited to the self-reported preference for writing which leads to three categories: "right hand", "left hand" and "both hands". These data present some peculiarities. The rate of ambidexterity is reported at ~1.5% in the population, which is higher than expected. In fact, individuals who can write equally well with both hands are extremely rare. The heritability estimates for left-handedness and ambidexterity are also puzzling. At behavioural level, the identification of siblings (N = 20,277 pairs) and other relatives (N = 49,788 pairs) led to a heritability ($h^2$) estimate of 12% for left-handedness [22]. This sample size was too small to derive a reliable estimate for ambidexterity with the same approach. Instead, genome-wide molecular data showed that common genetic variants, or single nucleotide polymorphisms (SNPs), capture up to 6% and 15% of the heritability ($h_g^2$) for left-handedness and ambidexterity, respectively. The higher $h_g^2$ observed for ambidexterity is a potentially exciting finding, but it also revealed some bias. Thanks to the molecular data, it is possible to test the genetic correlation across different traits. This analysis revealed that ambidexterity did not show genetic correlation with left-handedness or other neurodevelopmental, neural, and psychiatric traits as expected and as observed for left-handedness [22]. Instead, ambidexterity showed genetic correlation with the risk of being injured. A possibility is that the ambidexterity measure (reported as "being able to write with both hands") may be a consequence of injuries that force the use of the nonpreferred hand. However, to further complicate the situation a very recent genetic study for dyslexia—a neurodevelopmental phenotype—found a significant genetic correlation with the UK Biobank ambidexterity measure (but not with left-handedness) in over a million individuals derived from the 23andMe database [23]. The results of these studies show that even extremely large samples are not sufficient to disentangle patterns of associations between various binary traits and emphasise the importance of the quality of the phenotypes used for genetic studies.

More detailed handedness assessments were possible in smaller cohorts. The ALSPAC cohort is exceptional for the richness of measures collected over three decades. In addition to multiple measures, it also offers the advantage of a family structure design with both behavioural and genetic data available in parents and children. Taking advantage of these features, we were able to derive and compare heritability estimates across different handedness measures [24]. We found that $h_g^2$ for left-handedness, as a categorical measure, was 8%—a slightly higher but comparable figure to the 6% observed in the UK Biobank. When transforming the categorical phenotypes in quantitative scores, the $h_g^2$ for measures of hand preference derived from the summary of EHI scores was 21% (this transformation was achieved by regressing out effects of sex, age, and the first two principal components for ancestry as described by Verhoef et al. [25]). This 21% figure is similar to the estimates derived from behavioural analysis in twins. The same analysis for individual items showed variability across activities. For example, the highest heritability estimate (42%) was observed for the "hand used to cut" item. The same item presented the higher heritability (32%) also in a Japanese study [26]. These data both support the benefit of using quantitative phenotypic transformations and indicate that individual, rather than summary or composite measures, might be a more powerful tool to capture genetic factors underlying handedness. The same conclusion was reached by a study in a Mexican sample [27] and support the idea that different handedness measures capture different components of handedness. A key feature of quantitative phenotypes is that they distinguish both the poorly and the extremely lateralised individuals in addition to the left- and right-handedness direction (Figure 1). Therefore, if genetic factors contribute to the degree of lateralization rather than the direction of handedness, such effect will not be captured by individual measures of hand preference. When we applied the same phenotypic transformations to laterality measures other than handedness (i.e., foot and eye preference), we found that the heritability of foot preference was 28%—higher than what observed for handedness—but

was negligible for eye preference. This finding, in agreement with behavioural data from a previous study [28], suggests that other laterality measures beyond hand preference have the potential to lead to significant genetic discoveries.

However, the ideal scenario of having multiple handedness measures in large cohorts remain challenging. The "preferred hand for writing" is a very convenient way to assess handedness because it can simply be a box ticking as part of larger studies. For example, large cohorts primarily designed for studying the genetics of various diseases, can then be reanalyse for the genetics of hand preference at no extra cost. However, a hand preference measure might not capture genetic factors contributing to different aspects of handedness. In particular, it is not an ideal way to identify mixed-handed or poorly lateralised individuals. Currently, large scale collection on LQ measures is challenging and requires significant resource. For example, it would require dedicated personnel to collect, record and entry the data. As we move towards increasing digitalisation of every aspect of our lives, online platforms could offer a viable route for the collection of laterality data in large populations.

### 3. There Is No Handedness Gene

Having firmly established that a large generic genetic component underlies handedness, the next question is what specific genes determine whether one individual is right- or left-handed? Currently, our best answer is "many and not one in particular". It is now universally accepted that there is no single gene or single allele determining left handedness, contrary to what predicted by the theories proposed in the 80s [29,30]. However, it is important to recognise the values of these theories, which fitted with the data available at the time and played key roles in driving research efforts in the field. Thanks to recent advances in genomic technologies, we are now appreciating the highly polygenic nature of neurodevelopmental traits and of common human diseases. In fact, such complexity is much higher than it was anticipated only 10 years ago [31]. Genomic technologies include both genotyping of known variants, or SNPs, used for genome-wide association studies (GWAS) and resequencing to discover rare or *de novo* variants. Both technologies have generated data for hundreds of thousands of individuals and, considering that, as we just discussed, hand preference for writing is a very straightforward variable to be collected, a large amount of data for gene mapping are available. The fact that no specific individual gene with large effect was identified yet unequivocally excludes the possibility that there could be one single genetic factor causing left-handedness. Instead, an increasing number of genes with small effect sizes are being found in parallel with the analysis of increasingly large cohorts, confirming the polygenic nature of this trait (Table 1).

### 4. Resequencing the Genome

The ability to resequence the human genome at affordable cost allows the identification of genetic variants characterised with large effect sizes on the phenotype. Such effect tends to be disruptive and reduce the fitness of an individual. The large effect size is usually due to changes in the coding sequence that in turn alter the function of the corresponding protein. Whole exome sequencing (WES) technology targets specifically the coding regions (~2% of the entire genome) and offers an efficient way to discover such variants. Whole genome sequencing (WGS) instead covers the entire genome. Compared to that of WES, WGS is more expensive and poses the challenge of handling and interpreting a very large amount of data. It is estimated that each of us carry thousands of rare variants and up to 100 *de novo* mutations that are not inherited from our parents. Dissecting which ones might be relevant for the phenotype under investigation is not straightforward. WES has the advantages of being cheaper in terms of data generation and more straightforward in terms of data handling cost compared to that of WGS. The downside is that WES cannot detect potential functional mutations located in regions far from genes, e.g., regulatory enhancers, and is not ideal for the identification or larger insertions and duplications. While sample size remains a key factor for the interpretability of sequencing studies, the selection of participants is also vital. In general, the severity of the phenotype could be a good indicator

of the presence of causative rare variants. For example, WES studies are a powerful tool for the identification of mutations causing undiagnosed severe neurodevelopmental delays [32,33]. However, these are examples of clearly severe and debilitating phenotypes that cannot be compared directly with a left-handedness status, even in the case of "strong" or "extreme" left hand preference assessed with the EHI and LQ measures. A strategy that mirror this selection criteria was adopted by a WGS study that focussed on individuals with *situs inversus*, a left-right reversal of the visceral organs, and who presented an elevated rate left-handedness [34]. Although in a few cases mutations in genes known to contribute to laterality defects were detected, no obvious genetic causes were identified for five individuals with *situs inversus*, three of which were also left-handers. These data suggest that even extreme asymmetric phenotypes are not a specific category cause by single variants with large effects.

Another strategy for the selection of individuals in sequencing studies is to focus on families presenting a clear inheritance pattern suggestive of a mutation that co-segregates with the phenotype. This approach was successful in mapping genes underlying different traits and diseases including language-related disorders. Although the discovery of the *FOXP2* gene goes back to the pre-genomic era, it was due to the observation of a severe speech and language disorder in multiple members of a large multigenerational family, consistently with the presence of a dominant mutation [35]. Other mutations contributing to language impairment were identified either through the analysis of large families [36], as well as in individual cases selected for severity [37]. Such approaches demonstrate the power of sequencing studies in detecting single causative genetic factors also in the context of highly polygenic traits like language impairment. This scenario shows that polygenic traits can result from single mutations, however these mutations are likely to occur in different genes, and therefore, are difficult to detect. Causation can be inferred when the same variant or different variants in the same gene are observed in multiple individuals. Very few sequencing studies, conducted specifically to map genes causing left-handedness, were conducted so far. Two separate WES studies sequenced members of families that practiced consanguineous marriage and presented an overrepresentation of non-right-handed individuals. The assumption of the sequencing studies was that left-handers in these families would carry a causative genetic variant [38,39]. Neither of the studies found any compelling evidence that this was the case. While a causative mutation located outside the regions covered by WES cannot be completely ruled out, the most likely interpretation of these negative findings is to add support to the polygenic nature of handedness. Given the limited number of sequencing studies, we cannot reach definitive conclusions, and the identification of single mutations directly causing left-handedness remains a possibility. However, considering the evidence collected so far, we expect this scenario to be an exception rather than the rule.

## 5. Handedness GWAS

Increasingly large GWAS for handedness measures led to a growing number of statistically significant genetic associations (Table 1).

The most recent GWAS, and the largest to date (N = 1,766,671) conducted by Cuellar-Partida et al., for a categorical definition of handedness confirmed the highly polygenic nature of handedness [22]. Such an impressive sample size was reached by analysing study participants from the UK Biobank, 23andMe (https://www.23andme.com, accessed on 25 August 2021), and the International Handedness Consortium. The study identified 48 statistically significant associations, of which 41 were associations with left-handedness, and 7 with ambidexterity. In addition to the detection of these associations, what the study did not find is equally compelling. Firstly, there was no single genetic factor associated with a large effect and, second, it is clear that many other genetic factors, beyond these 48 associations, remain to be identified. Overall, these observations confirm the highly polygenic nature of handedness, which is expected to implicate a much larger pool of genes than the ~40 genes predicted by McManus and colleagues in 2013 [40]. Cuellar–Partida et al.

conducted multiple analyses in addition to individual marker-traits associations providing new insights into the biological pathways contributing to handedness [22]. They observed that the genetic correlation between left-handedness and ambidexterity was very low. As discussed above, this is probably explained by a bias introduced by the self-reported measures, which are do not capture genuine ambidexterity. Instead, tissue- and pathway-enrichment found that genetic associations for left-handedness (but not ambidexterity) suggested, as expected, a role of the central nervous system. In particular, left-handedness was associated with genes involved in the activity or formation of microtubules, including *MAP2*, *TUBB*, *TUBB3*, *NDRG1*, *TUBB4A*, *TUBA1B*, *BUB3*, and *TTC28*. Microtubules are major components of the cytoskeleton and are essential for many processes, such as cell division, cell motility, intracellular transport, and maintenance of cell shape. Increasing evidence is supporting the role of microtubules in neurodevelopment and neurodevelopmental disorders [41,42]. Given the association between handedness and some psychiatric conditions, e.g., schizophrenia [43], Cuellar–Partida and colleagues suggested that microtubule-mediated processes could mediate the link between asymmetries and disorders. Microtubules were proposed as a key element to explain this complex link by Wiberg and colleagues in an earlier GWAS conducted in a subset of the UK Biobank individuals (N~400,000) [44]. The findings from our previous GWAS for a LQ derived from the pegboard task, and conducted in a much smaller sample (N = 728), proposed that shared biological pathways contributing to the establishment of left/right anatomical differences would also contribute to handedness and brain asymmetries [45]. Specifically, we suggested cilia-mediated processes as one of these biological pathways [46–48]. Cilia are microtubule-based cellular structures with sensory and motility function. During early development, cilia are critical in pattering the left/right axis determination and mutations in genes controlling cilia formation and function lead to laterality defects (See Vingerhoets et al. in this issue for a detailed explanations of this biological pathways [10]). The specific marker-trait associations from our study did not replicate in the larger GWAS for categorical measures of handedness, and it is possible that the lack of replication is due to the limited power of the original study, which led to false positives. Alternatively, it is possible that the different results are explained by the use of a quantitative LQ versus categorical phenotypes. As discussed earlier, the LQ measure different handedness dimensions better suited to capture the underlying genetic component. Beyond the individual associations, microtubules functions and formation (e.g., cilia and cytoskeleton dynamics) are unifying themes across the different studies. Together, the molecular genetics studies support the polygenic nature of handedness (Figure 1), disproving the single-gene theories and suggesting a scenario more in line with the liability threshold model [49]. This model proposed that binary traits are the results of multiple factors, each contributing a small effect, and normally distributed in the population. A threshold along the liability distribution determines the status of an individual for one of the two trait categories.

## 6. Genetics, Handedness and Brain Asymmetries

The first link between handedness and brain imaging genetics was suggested by Wiberg and colleagues in their GWAS [44]. One of the top associations with handedness, i.e., the rs199512 SNP located in the *WNT3* gene, was also associated with measures of white matter structural connectivity in brain regions involved in language, including the tracts linking Broca's and temporoparietal junction areas. A limitation of this analysis was the use of a single marker. Instead, a feature of large GWAS is that they allow the generation of polygenic risk scores (PRS), which capture the cumulative effect of associated variants [50]. It is then possible to test whether PRS for a particular trait derived from large GWAS (the training sample) influence other traits in separate samples which can be small in size (the target sample). For example, PRSs for educational attainment were among the first to become available [51] and were derived from increasingly large cohorts of up to 1.1 million individuals [52]. PRSs for educational attainment were tested for association

with different cognitive, behavioural, and clinical traits, and were shown to account for about 2.1% of the variance in measures of reading abilities and dyslexia [53–55].

Under this principle, PRS for categorical measures of handedness, derived from a subset of UK Biobank participants (N = 331,037) [56], were tested by Ocklenburg and colleagues in a cohort of N = 296 participants [57]. They found that the PRS for hand preference were associated with LQ, showing the potential advantages of quantitative measures of handedness to capture genetic effects in samples of a modest size. Instead, no associations were detected with the brain measures selected for this study that focussed specifically on asymmetries in grey matter macrostructures.

Mapping genetic variants to functional and anatomical brain data is extremely challenging because of the large number of tests required by these analyses and high heterogeneity of the methods used in different studies. The ENIGMA (Enhancing Neuro-Imaging Genetics through Meta-Analysis; http://enigma.ini.usc.edu/) (accessed on 25 August 2021) consortium provides a platform to address these challenges and includes a working group focussed on brain laterality.

A number of studies looked for brain markers that could correlate with handedness measures. Brain imaging data in the UK Biobank provided evidence for associations between handedness and the overall anatomical hemispheric twist, or "torque" [58], and differences in functional connectivity in the language-associated regions in both hemispheres [44]. The next question is to ask whether associations between handedness and brain asymmetries could be mediated by shared genetics. In a very recent study, Sha and colleagues assessed the relationship between handedness and cortical asymmetries by generating asymmetry maps for cortical thickness and surface area in 28,802 right-handed and 3062 left-handed UK Biobank participants [59]. They found several regions that differed between left- and right-handers, consistent with a shift of neuronal resources to the hemisphere controlling the dominant hand. This means a general less leftward/more rightward shift for left-handers, who have a right hemisphere dominance for the preferred hand. Next, the same study derived PRS for handedness in an independent training sample of individuals from the UK Biobank to be tested in the target sample of individuals selected for the initial brain imaging analysis. As expected, the PRS were associated with handedness in the target sample. However, the handedness PRS also showed associations with cortical surface area asymmetries that differed between left- and right-handers. Specifically, PRS increasing the chances of left-handedness were associated with increased average rightward asymmetry in the fusiform cluster and decreased average leftward asymmetry in the anterior insula clusters. Tubulin-associated genes featured among the genes associated with cortical asymmetries. This is not surprising considering that these types of genes were enriched in the associations with handedness.

**Table 1.** GWAS for handedness measures.

| Reference | N Participants | Cohorts | Handedness Phenotype | N Associated Genes |
|---|---|---|---|---|
| Eriksson et al. 2010 [60] | 9126 | 23andMe | Handedness questionnaire | none |
| Scerri et al. 2011 [61] | 744 | Dyslexia cohorts and ALSPAC | LQ from pegboard task | 1 |
| Brandler et al. 2013 [45] | 728 + 2666 | Dyslexia cohorts and ALSPAC | LQ from pegboard task | 1 |
| Wiberg et al. 2019 [44] | ~400,000 | UK Biobank | Hand preference | 4 |
| De Kovel et al. 2019 [56] | 331,037 | UK Biobank | Hand preference | 3 |
| Cuellar–Partida et al. 2021 [22] | 1,766,671 | UK Biobank, 23andMe, International Handedness Consortium | Hand preference | 48 |

These studies illustrate the challenges of conducting these types of analyses, which require large samples and rigorous methodology. Resources like the UK Biobank are a real gamechanger for this field. The large sample size allows detecting subtle effects of genes associated to complex phenotypes. These findings are the initial step to start disentangling at molecular level the relationship between handedness and cerebral asymmetries.

## 7. Conclusions

The two critical elements for the success of genetic studies are the sample size and the quality of the phenotype. Resources such as the UK Biobank demonstrate how large sample sizes allow the detection of subtle effects, as well as linking different types of data collected in relatively homogeneous ways across many individuals. Such studies led to the identification of specific genes associated to hand preference, implicating specific biological pathways, such as the function and formation of microtubules, to be relevant to both handedness and cerebral asymmetries. These discoveries relied on the use of the preferred hand for writing as handedness phenotype. This is a very convenient measure for the collection of large-scale data. However, these discoveries explain only a tiny fraction of the genetics contributing to handedness, and many more genes remain to be identified. While even larger samples characterised with hand preference measures will probably lead to the discovery of additional genes, the use of different types of handedness measures could provide another valid route for gene discovery. The modest correlation across handedness measures indicates that each of them captures a distinct dimension of handedness. Some of these measures also present heritability estimates that are higher than those observed for categorical measures of hand preference, and therefore, are more suited for genetic studies. In an ideal scenario, multiple handedness measures collected in large samples are likely to lead to novel breakthroughs. With the increased level of digitalisation and online testing [62], these types of datasets are becoming a more likely and extremely exciting possibility. For now, one of the key advances in the field is a new appreciation for the complexity that underlies handedness, a trait apparently very simple at both the behavioural and molecular level.

**Funding:** This work received no external funding. SP is supported by the Royal Society [UF150663]. The APC was funded by The University of St Andrews.

**Conflicts of Interest:** The author declares no conflict of interest.

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
