# Peer review of "Recent Advances in Handedness Genetics"

_symmetry, doi:10.3390/sym13101792_

Round 1

Reviewer 1 Report

The manuscript of Silvia Paracchini is a nice review of recent works about handedness genetics. It is well written, well structured from results of twin studies to GWAS, pointing to the polygenic nature of handedness and multidimensional nature of handedness.

My main comment is that despite the author points to the discrepancies between the discrete phenotype of "the prefered hand for writing" and the quantitative nature of laterality, she doesn't develop about the notion of threshold trait and liability trait (ie the  continuously distributed underlying trait). They are large amount of quantitative genetics theory about such traits and their polygenic nature despite being discrete which seems to me a bit relevant here.

Many works of Michel Raymond (Univ. Montpellier) discuss evolutionary scenario such as balancing selection for the maintenance of handedness. This could point the author for some literature references when she discuss evolutionary forces.

I think "causative mutations located outside genes, which are expected to very improbable" is a bit misleading on what the author thought of that. This is an assumption of WES and not a very likely biological assumption for laterality based on what I've read before (multidimensional, polygenic, with a large stochastic component linked to developmental processes and thus to regulatory elements). Reformulating the sentence will certainly clarified this point.

Minor comments

p1. As such, the first sentence sounds very useless.

p3. The sentence starting with "Tools like the Edinburg ..." is repeated.

Author Response

The manuscript of Silvia Paracchini is a nice review of recent works about handedness genetics. It is well written, well structured from results of twin studies to GWAS, pointing to the polygenic nature of handedness and multidimensional nature of handedness.

->Thank you.

My main comment is that despite the author points to the discrepancies between the discrete phenotype of "the prefered hand for writing" and the quantitative nature of laterality, she doesn't develop about the notion of threshold trait and liability trait (ie the  continuously distributed underlying trait). They are large amount of quantitative genetics theory about such traits and their polygenic nature despite being discrete which seems to me a bit relevant here.

-> I have added some considerations on this point and a new figure to illustrate how a quantitative measure of handedness might better capture the polygenic component.  

Many works of Michel Raymond (Univ. Montpellier) discuss evolutionary scenario such as balancing selection for the maintenance of handedness. This could point the author for some literature references when she discuss evolutionary forces.

-> Great suggestion. I have added additional references in the relevant section.

I think "causative mutations located outside genes, which are expected to very improbable" is a bit misleading on what the author thought of that. This is an assumption of WES and not a very likely biological assumption for laterality based on what I've read before (multidimensional, polygenic, with a large stochastic component linked to developmental processes and thus to regulatory elements). Reformulating the sentence will certainly clarified this point.

-> After re-reading the sentence, I agree it was not clear. This paragraph is mainly aimed at distinguishing the benefits of WES versus WGS, and explaining while it is still the sequencing approach of choice. I rewrote it to clarify this point and add a sentence at the end of the paragraph to spell out the possible (but unlikely failure of WES to pick up a causative variants outside the covered region. Also related to the point made by the reviewer, I use the example of language impairment to show that a polygenic trait can be cause by individual mutations in isolated cases. We cannot exclude such individual causative mutations for handedness, but they are expected to be rare. I added a sentence to spell out this point more directly and specified whether the studies mentioned used WES versus WGS.

Minor comments

p1. As such, the first sentence sounds very useless.

-> I am not sure to what sentence the reviewer is referring to. Is it the first sentence in the abstract (pag 1) or of the introduction (pag 2). I have edited the first sentence of the introduction as part of the general revision.

p3. The sentence starting with "Tools like the Edinburg ..." is repeated.

-> Yes! Thank you for spotting that. Duplication now fixed. This was an artefact created by OneDrive which duplicated different sections. I went through the document to check that all duplications were removed

Reviewer 2 Report

Overall, the review is good and detailed. The author substantiates his assumptions with references to articles from resensible journals.
But I absolutely cannot agree with the categorical phrase on page 2: "The link between handedness and language, which is also strongly lateralised and presenting left- hemispheric dominance in most people, is another indicator of the biological nature of handedness (5)"
It is now well known that language skills are provided by the coordinated work of many brain structures. And it is often said about dominance in different paired brain structures when solving different linguistic tasks. For example, fMRI data show an increase in oxyhemoglobin levels and a decrease in deoxyhemoglobin levels in the left inferior frontal lobe, including Broca's area (Quaresima et al., 2002), when performing a translation task and a language-switching task.  In the switching task, the N2 component of the ERP in the left frontal-central area is more negative compared to the task without switching (Crinionetal., 2006).  Meanwhile, various studies assign a key role in language switching in bilinguals to either the left caudate nucleus (Crinionetal., 2006) or the right caudate nucleus (Wangaetal., 2007). Also, a short article "Behavioral and cognitive correlates of foreign language proficiency" (https://doi.org/10.1016/j.ijpsycho.2014.08.840) highlights results indicating the importance of brain asymmetries of different types in solving linguistic tasks at different ages. Thus, it is definitely not possible to speak of "strict lateralization". 
Moreover, it is worth mentioning that brain asymmetry exists of different types, and functional hemispheric asymmetry must not be forgotten. I suggest the author add information about this, referring for instance to the article "Does Double Biofeedback Affect Functional Hemispheric Asymmetry and Activity? A Pilot Study" ( Symmetry 2021, 13(6), 937; https://doi.org/10.3390/sym13060937 ). There is a rationale for distinguishing between types of brain asymmetry, as well as a description of the principle of measuring functional interhemispheric asymmetry.

Author Response

Overall, the review is good and detailed. The author substantiates his assumptions with references to articles from resensible journals.

-> Thank you!

But I absolutely cannot agree with the categorical phrase on page 2: "The link between handedness and language, which is also strongly lateralised and presenting left- hemispheric dominance in most people, is another indicator of the biological nature of handedness (5)"

It is now well known that language skills are provided by the coordinated work of many brain structures. And it is often said about dominance in different paired brain structures when solving different linguistic tasks. For example, fMRI data show an increase in oxyhemoglobin levels and a decrease in deoxyhemoglobin levels in the left inferior frontal lobe, including Broca's area (Quaresima et al., 2002), when performing a translation task and a language-switching task.  In the switching task, the N2 component of the ERP in the left frontal-central area is more negative compared to the task without switching (Crinionetal., 2006).  Meanwhile, various studies assign a key role in language switching in bilinguals to either the left caudate nucleus (Crinionetal., 2006) or the right caudate nucleus (Wangaetal., 2007). Also, a short article "Behavioral and cognitive correlates of foreign language proficiency" (https://doi.org/10.1016/j.ijpsycho.2014.08.840) highlights results indicating the importance of brain asymmetries of different types in solving linguistic tasks at different ages. Thus, it is definitely not possible to speak of "strict lateralization". 
Moreover, it is worth mentioning that brain asymmetry exists of different types, and functional hemispheric asymmetry must not be forgotten. I suggest the author add information about this, referring for instance to the article "Does Double Biofeedback Affect Functional Hemispheric Asymmetry and Activity? A Pilot Study" ( Symmetry 2021, 13(6), 937; https://doi.org/10.3390/sym13060937 ). There is a rationale for distinguishing between types of brain asymmetry, as well as a description of the principle of measuring functional interhemispheric asymmetry.

-> While I agree with the reviewer that it is important to recognise the role of both hemispheres during language tasks, it is well established that one hemisphere – the left one in most people – is dominant for language processing. I edited the relevant section to be more precise. Hopefully, the reviewer noted that in my final section I refer specifically to “differences in functional connectivity in the language-associated regions in BOTH hemispheres”. I now spell out in the same sentence that some asymmetries can be “anatomical”.  Instead, as this review focusses on handedness rather than language, I have not included the above references because I thought it would diverge the narrative. Instead, I referred to two other reviews in this same special issue covering this topic in more details.

I am glad to see that a part from this sentence, the rest of the review did not raise other issues.

Round 2

Reviewer 2 Report

Author should carefully look through the typos:

Right hemisphere dominance for language is rare and observed preferentially in let-handers 

And of course edit the paper according to the requirements.